# The Adsorption Behavior of Gas Molecules on Co/N Co–Doped Graphene

**DOI:** 10.3390/molecules26247700

**Published:** 2021-12-20

**Authors:** Tingyue Xie, Ping Wang, Cuifeng Tian, Guozheng Zhao, Jianfeng Jia, Chenxu Zhao, Haishun Wu

**Affiliations:** 1Key Laboratory of Magnetic Molecules & Magnetic Information Materials Ministry of Education, School of Chemistry and Material Science, Shanxi Normal University, Taiyuan 030006, China; tingyuexie@126.com (T.X.); zhaoguozheng@sxnu.edu.cn (G.Z.); jiajf@dns.sxnu.edu.cn (J.J.); 2School of Physical and Electronics Science, Shanxi Datong University, Datong 037009, China; wangping061226@aliyun.com (P.W.); cftian_050@sxdtdx.edu.cn (C.T.); 3Institute of Environmental and Energy Catalysis, School of Materials Science and Chemical Engineering, Xi’an Technological University, Xi’an 710021, China

**Keywords:** density functional theory (DFT), electronic properties, gas adsorption

## Abstract

Herein, we have used density functional theory (DFT) to investigate the adsorption behavior of gas molecules on Co/N_3_ co–doped graphene (Co/N_3_–gra). We have investigated the geometric stability, electric properties, and magnetic properties comprehensively upon the interaction between Co/N_3_–gra and gas molecules. The binding energy of Co is −5.13 eV, which is big enough for application in gas adsorption. For the adsorption of C_2_H_4_, CO, NO_2_, and SO_2_ on Co/N–gra, the molecules may act as donors or acceptors of electrons, which can lead to charge transfer (range from 0.38 to 0.7 e) and eventually change the conductivity of Co/N–gra. The CO adsorbed Co/N_3_–gra complex exhibits a semiconductor property and the NO_2_/SO_2_ adsorption can regulate the magnetic properties of Co/N_3_–gra. Moreover, the Co/N_3_–gra system can be applied as a gas sensor of CO and SO_2_ with high stability. Thus, we assume that our results can pave the way for the further study of gas sensor and spintronic devices.

## 1. Introduction

In recent years, many carbon allotropes, such as graphene and graphite, have served as a broadly and highly capable material due to its perfect properties in various aspects including electricity, chemistry, mechanistic, thermology, and optics [1,2]. Among these candidates, graphene is the simplest allotrope and has been extensively syntehsized in experimental works. Further, graphene is a typical two dimensional material which possesses large surface area. This character can lead to perfect adsorption ability for some gas molecules. Thus, graphene is a promising candidate for the detection and capture of gas molecules [3]. Upon the adsorption and desorption of gas on graphene, the conductivity of the complex will experience a dramatic change, which is the origin of its reactivity for gas sensing [4]. As such, the application of graphene derivative in areas of gas sensors and spintronic devices has attracted increasing attention [5,6,7]. For example, the electric property of reduced graphene oxide can change sensitively with the surface adsorption, leading to extensive investigation and application in the area of gas sensing [8].

Pristine graphene always interacts with molecules weakly via Van der Waals force, leading to less charge transfer. Thus, graphene exhibits no reactivity for the detection of CO and H_2_O [9]. In order to improve the sensitivity of graphene for gas sensing, introducing a single atom catalyst (SAC) into graphene substrate is a promising approach which can increase the binding strength of gas molecules dramatically [10]. This can also change the electric properties of graphene systems simultaneously [11,12,13,14,15,16]. This type of configuration has been widely investigated in pioneering works: Zhao et al. have embeded Ni atoms on different graphene–based materials as a novel material for the detection of acid gases [14]. Yang et al. have also proposed a Co–N_3_ decorated graphene and applied it into CO oxidation [15]. Metal decoration can effectively enhance the adsorption strength of gas molecules: Co (Li)–decorated graphene can capture NO (CO) with −4.04 eV (−0.55 eV) adsorption energy [16]. However, the high surface energy of doped graphene may lead to the aggregation of atoms, leading to the stability issues. Among numerous dopants, Co has been proved stable in graphene’s mono– or double–vacancies [17]. Many methods have also been carried out to introduce vacancies in graphene, such as particle beam bombardment, chemical vapor deposition, and triggering defects via crystal growth [18,19]. These methods can regulate the doping process of transition metals (TMs) into graphene effectively. Moreover, graphene doped with three N atoms (N_3_–gra) has been successfully synthesized in experimental works, which can change the electric properties of graphene and possesses extensive applications [17,20,21,22,23,24]. Chen et al. have discovered the potential application of graphene as oxygen reduction catalyst and proposed heteroatom doping as an excellent method to tune the catalytic reactivity [20]. Lukaszewicz et al. have also synthesized a micro–mesoporous graphene and highlighted the structural factors in application of Zn–air batteries [21]. In addition, metal dopants can also change the magnetic properties of graphene via the regulation of initial spin states near the Fermi level [25,26,27,28,29]. Kettel et al. have successfully constructed TM/Nx co–doped graphene and proved this configuration manifesting high stability [21]. The electric and magnetic properties have also been calculated and investigated deeply by Kettel et al. However, the change of properties under the influence of gas adsorption still lacks detailed investigation.

Herein, we have performed density functional methods to investigate the adsorption of gases, including C_2_H_2_, CO, NO_2_, and SO_2_, on Co/N_3_ co–doped graphene (Co/N_3_–gra). In addition, we have also calculated the change of properties after gas adsorption including geometric stability, electronic structure, and magnetic properties. Our findings have not only clarified the mechanism of interaction between gas molecules and Co/N_3_–gra but also paved the way for the application of graphene–based materials in gas sensors.

## 2. Computational Details and Methods

All calculations are performed within the Vienna ab–initio simulation package (VASP), which is based on density functional theory (DFT) and the projector augmented wave (PAW) pseudopotentials method [30,31,32]. The approximate deviation of DFT local functionals are 0.2 eV for energy calculation, indicating the validation of DFT method in gas adsorption area. The generalized gradient approximation (GGA) with the Perdew Burke Ernzerhof (PBE) functional is used to describe the exchange and correlation potential [33]. The interaction of the van der Waals (vdW) is considered with DFT–D3 [34,35] to improve the expression of weak interaction between gas molecules and substrates. Charge transfer was calculated with the Bader charge analysis method [36]. In our work, we use a hexagonal supercell (4 × 4 × 1) with periodic boundary conditions. A vacuum spacing of 15 Å is used to model the infinite graphene sheet, which is sufficiently large to obtain reliable results [25,37]. A plane wave energy cutoff was chosen to be 500 eV, and the width of the Gaussian broadening scheme is σ = 0.05 eV occupation of electron level [38]. In the geometric optimization process, the position relaxation of all atoms is calculated by conjugate gradient method until the maximum force is smaller than 0.01 eV/Å. The total energy convergence accuracy is 10^–6^ eV for the electronic self–consistent steps. In order to improve the accuracy of calculation and save time simultaneously, a k–point grid with 7 × 7 × 1 as the center is used to sample the Brillouin region for structural optimization. The k–point mesh of 15 × 15 × 1 is used to calculate the density of state (DOS). All the configurations are visualized via Vesta [39].

The binding energy of single Co atom (*E*_b_[Co]) in Co/N_3_–gra is expressed as:*E*_b_[Co] = *E*[Co] + *E*[v–Co/N_3_–gra] − *E*[Co/N_3_–gra]
where the *E*[Co/N_3_–gra], *E*[v–Co/N_3_–gra], and *E*[Co] denote total energy of Co/N_3_–gra, Co/N_3_–gra with the Co atom moved away, leaving as Co vacancy (v–Co) and single Co atom in vacuum, respectively.

The adsorption energy of gas molecules *E*_ad_[gas] on Co/N_3_–gra is expressed as:*E*_ad_[gas] = *E*[gas] + *E*[Co/N_3_–gra] − *E*[gas–Co/N_3_–gra]
where the *E*[gas–Co/N_3_–gra], *E*[Co/N_3_–gra], and *E*[gas] denote total energy of gas adsorbed Co/N_3_–gra, Co/N_3_–gra surface, and single gas molecules in vacuum, respectively. The configurations we constructed are all periodic structures and the size of unit cell are set big enough to avoid the influence of Vander Waals force between molecules.

## 3. Results and Discussion

### 3.1. The Configuration and Stability of Co/N_3_–Gra

We first study the configuration of Co/N_3_–gra substrate. The final structure after full optimization is shown in Figure 1. The Co atom, located at the center of defective site, can bind with three N atoms equally with bond length of 1.43 Å, which agrees with pioneering study [37]. In addition to bond length, the binding energy of Co atom (*E*_b_[Co] = −5.13 eV) also exhibits similar value to pioneering result (*E*_b_[Co] = −4.80 eV), indicating the accuracy of our calculation [37]. The *E*_b_[Co] calculated is much larger than the cohesive energy of Co atom (−4.43 eV), which is beneficial for prohibiting the aggregation of Co into clusters [40]. Overall, the Co/N_3_–gra complex possesses high geometric stability. Compared to graphene with a single vacancy (v–gra), Co can transfer a larger number of electrons to N_3_–gra substrate (0.80 e vs. 0.68 e) based on analysis of Bader charge [25]. Further, Co interacts with N_3_–gra via covalent bonds, reflected from the significant overlap of charge density between Co and N atoms (see Figure 1). These characters are beneficial for Co/N_3_–gra to adsorb gas molecules.

We now focus on the adsorption of various gas molecules on Co/N_3_–gra. The molecules are located on the substrate with different orientations to confirm the optimal adsorption manner. The favorable configurations after full optimization are displayed in Appendix A in Appendix A. C_2_H_2_ located above Co atom with a distance of 1.91 Å and the C–C bond (H atoms) located parallel to graphene plane (raised upwards). There exists 0.38 e charge transfer from substrate to C_2_H_2_, leading to an adsorption energy of *E*_ad_[C_2_H_2_] = 1.95 eV (Appendix A in Appendix A). CO binds with Co site in Co/N_3_–gra via C atom, leading to an adsorption energy (distance) of *E*_ad_[CO] = −2.27 eV (1.76 Å). The result is favorable than that on v–gra (*E*_ad_[CO] = −0.62 eV). Thus, CO can be stabilized with the influence of N doping in Co/N_3_–gra. This can also be reflected from the charge transfer, which is 0.31 e between CO and Co/N_3_–gra but only 0.23 e for v–gra [25] (Appendix A in Appendix A). For NO_2_ and SO_2_, these molecules tend to bind Co with the O atoms due to the electron–rich character of O, which is similar to that on Pt/Pd–doped v–gra [27,41]. NO_2_ and SO_2_ show similar charge transfer (0.70 e) and adsorption distance (1.93 Å) but distinct adsorption energies (*E*_ad_[NO_2_] = −3.32 eV; *E*_ad_[SO_2_] = −1.85 eV). This can be interpreted from the change of configuration caused by charge redistribution: The average distance of Co–N (average bond length of NO_2_) has enlarged by 3.28% (10.34%) after NO_2_ adsorption on C/N_3_–gra. Further, N doping can also enhance the binding strength of SO_2_ like CO: The adsorption energy of SO_2_ is only −1.07 eV on Co–v–gra (compared to −1.85 eV on Co/N_3_–gra). Above all, CO and C_2_H_2_ bind with Co–v–gra via C atoms while it is O atoms for NO_2_ and SO_2_. NO_2_ and SO_2_ exhibit enhanced binding strength compared to CO and C_2_H_2_ due to large charge transfer.

### 3.2. Electric and Magnetic Properties

In order to understand the stability of gas molecules on Co/N_3_–gra, we focus our research on electric and magnetic properties. Electrons accumulate on molecules of C_2_H_2_, CO, NO_2_, and SO_2_, indicating an acceptor nature for gas molecules (Figure 2). This tendency is surprisingly notable for NO_2_ and SO_2_, indicating the electron environment has changed dramatically around adsorption sites. CO can alter its conductivity dramatically and C_2_H_2_ mainly changes its magnetic properties. The mechanism is discussed in detail as follows.

In pioneering works, co–doped v–gra has been proved as a magnetic material with a magnetic moment of 1.0 μB, which will increase to 2.24 μB after N doping [21,24,42]. As shown in the total DOS (TDOS), the Co/N_3_–gra complex exhibits semi–metal character due to the strong interaction between Co and N atoms (Appendix A). The peaks of up spin for Co/N_3_–gra are nearly identical to Co–3d states, which are located at −0.5 eV, −1.0 eV, and −2.5 eV. For states in down spin, The main hybridizations exist in the Co–3d and N–2p near the Fermi level, which is much stronger than that in up spin. This can be treated as the origin of strong interaction between Co and N atoms and enhanced magnetic moment before N doping.

Moreover, Co doping can also influence the C_2_H_2_ adsorption on Co/N_3_–gra. We thus calculate the DOS of C_2_H_2_ adsorbed Co/N_3_–gra (C_2_H_2_–Co/N_3_–gra) and compare it with bare Co/N_3_–gra. Co–3d states have experienced a broaden and decreased tendency after hybridization with Co–3d, indicating a strong bond strength between Co and C_2_H_2_ (Figure 3a). The decrease of TDOS near the Fermi level can also illustrate there exist large number of charge transfer between Co/N_3_–gra and C_2_H_2_. This phenomenon can also illustrate that the conductivity of Co/N_3_–gra can be modulated sensitively via C_2_H_2_ adsorption. The hybridization between C_2_H_2_ and Co–3d can also be made asymmetric between up and down spin for the TDOS near the Fermi level, leading to a magnetic moment increase of 0.24 μB.

For CO adsorption, the substrate can transfer 0.31 e charge to CO, leading to the change of DOS near the Fermi level (Figure 3b). Similar to C_2_H_2_ adsorption, the Co–3d states can also overlap with states of CO due to the strong bond strength with Co. The change of magnetic moment is 0.24 μB for CO adsorption, which is similar to C_2_H_2_ situation. However, the interaction between CO and Co/N_3_–gra can induce a semimetal to semiconductor transformation, which is particularly interesting.

For NO_2_ and SO_2_, the magnetic moment has decreased after adsorption on Co/N_3_–gra, which is different from that for C_2_H_2_ and CO. This may be due to the paramagnetic (nonmagnetic) property for NO_2_ (SO_2_) while the magnetic moment of Co/N_3_–gra is 2.0 μB in comparison [43]. The relatively low magnetism of NO_2_ and SO_2_ can induce a simultaneous shift of Co/N_3_–gra’s states in up and down spin towards the Fermi level, which eventually decreases the magnetic moment to 0.14~1.0 μB after adsorption of gas (Figure 3c). The peaks in TDOS of Co/N_3_–gra decrease after NO_2_/SO_2_ adsorption. The high interaction strength between NO_2_ and Co/N_3_–gra originate from the larger charge transfer of 0.70 e from Co to NO_2_ compared to that from Co to N atoms (0.35 e). This can be reflected from the overlap between Co–3d states and NO_2_ near the Fermi level. Compared to the adsorption of C_2_H_2_ and CO, NO_2_ molecule exhibits strong states overlap in the energy range of −2~−4 eV, which contributes significantly to adsorption and can be treated as the origin of high adsorption strength. Similar character can also be found in the SO_2_ situation (Figure 3d). Overall, NO_2_ and SO_2_ can regulate electronic and magnetic properties more effectively than C_2_H_2_ and CO via hybridization with Co–3d states. The changed properties induced by gas adsorption can be applied into spintronic devices. Among the gases investigated, the adsorption strength of C_2_H_2_ and SO_2_ is moderate and it is too strong for NO_2_. Thus, NO_2_ is likely to accumulate on Co/N_3_–gra and blocks the active sites, which will make Co/N_3_–gra different to recover in practical application. Thus, Co and N are promising dopants in graphene in the detection of C_2_H_2_ and SO_2_.

In order to clarify the spin character, we have also calculated the spin density of gas adsorbed Co/N_3_–gra (Figure 4). Different spin distributions are displayed for gases of C_2_H_2_, CO, NO_2_, and SO_2_. For C_2_H_2_/CO adsorption, the spin mainly concentrates on Co and N atoms and the character of the system is up spin (Figure 4a,b). For NO_2_ adsorption, the up spin mainly concentrates on N and Co atoms and O exhibits down spin. The up spin is dominant to down spin in general for NO_2_ adsorbed system. For SO_2_ adsorption, the SO_2_ exhibits down spin, which is opposite to Co/N_3_–gra substrate (up spin) (Figure 4d). Thus, the different adsorption systems can be classified based on spin distribution.

For comparison, we have also calculated the CO_2_ adsorption. The CO_2_ exhibits phyiscal adsorption on Co/N_3_–gra with an adsorption energy of −0.28 eV (Figure 5a,b). The bond angle of adsorbed CO_2_ maintains 180° as the free state. In addition, there exists no spin on CO_2_ adsorbed on Co/N_3_–gra (Figure 5c), indicating the adsorption of CO_2_ has nearly no change for the properties of the substrate. These phenomena mainly originate from the neglegible charge transfer between CO_2_ and Co/N_3_–gra (only 0.08 e).

## 4. Conclusions

In this study, we have performed density functional theory to study the adsorption of gas molecules, including C_2_H_2_, CO, NO_2_, and SO_2_, on Co/N_3_–gra substrate. Co/N_3_–gra possesses high stability due to the charge transfer from Co to N atoms. CO adsorption is particularly interesting, as it can realize the transformation of Co/N_3_–gra from semimetal to semiconductor. The adsorption of gases can regulate the electric and magnetic properties of systems, especially for NO_2_ and SO_2_, which can change the magnetism of the systems dramatically. Co/N_3_–gra can be treated as a promising sensor for the detection of C_2_H_2_ and SO_2_ due to its moderate adsorption strength. Our study has clarified the mechanism of gas adsorption in the aspects of electric and magnetic properties and paved the way for the application of Co/N_3_–gra in spintronic devices.

## Figures and Tables

**Figure 1 molecules-26-07700-f001:**
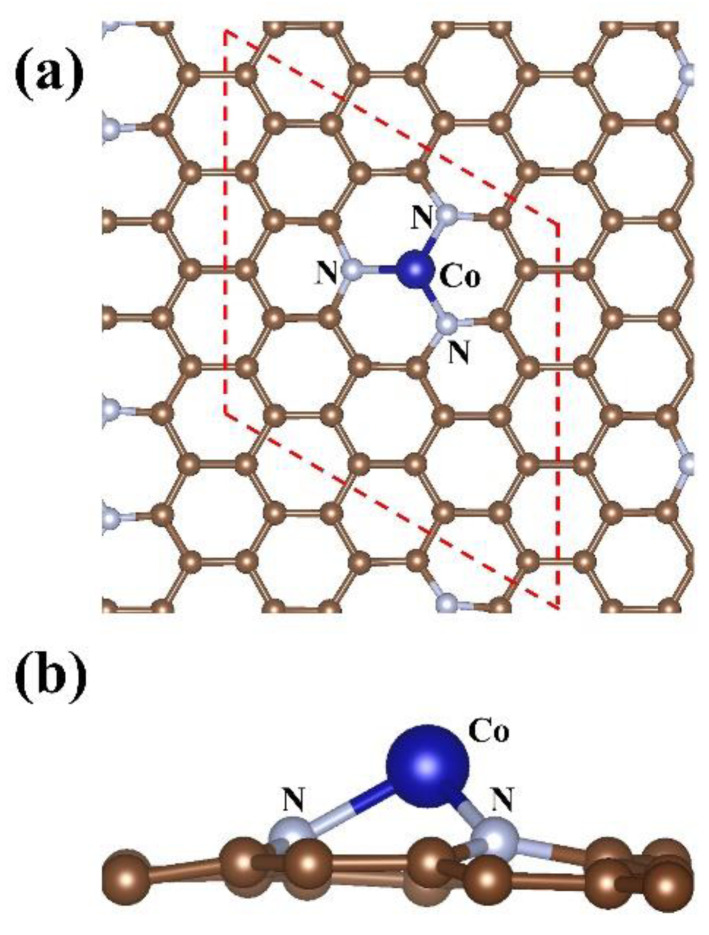
The (**a**) top view and (**b**) side view of fully optimized configuration of Co/N_3_–gra. The red dotted line in (**a**) denotes the unit cell.

**Figure 2 molecules-26-07700-f002:**
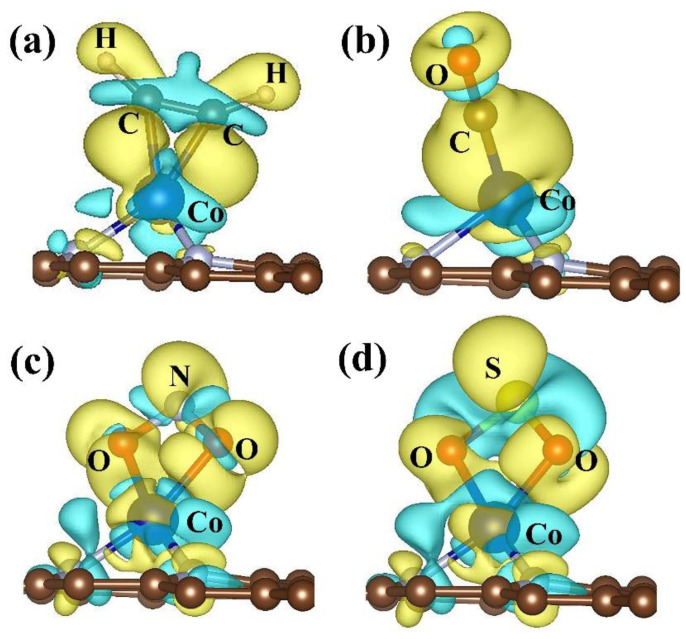
The charge density difference of Co/N_3_–gra with (**a**) C_2_H_2_, (**b**) CO, (**c**) NO_2_, and (**d**) SO_2_ adsorbed on it. The yellow and blue areas indicate the accumulation and depletion of charge, respectively. Isosurface value: 0.002 e/Bohr^3^.

**Figure 3 molecules-26-07700-f003:**
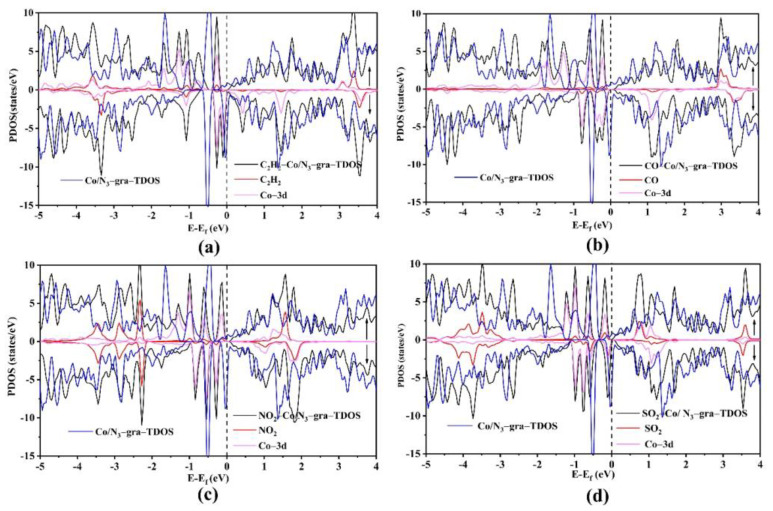
The density of states of gas molecules, including C_2_H_2_ (**a**), CO (**b**), NO_2_ (**c**), and SO_2_ (**d**), adsorbed Co/N_3_–gra, with the up (down) spin denoted as ↑ (↓). The dotted line indicates the Fermi level.

**Figure 4 molecules-26-07700-f004:**
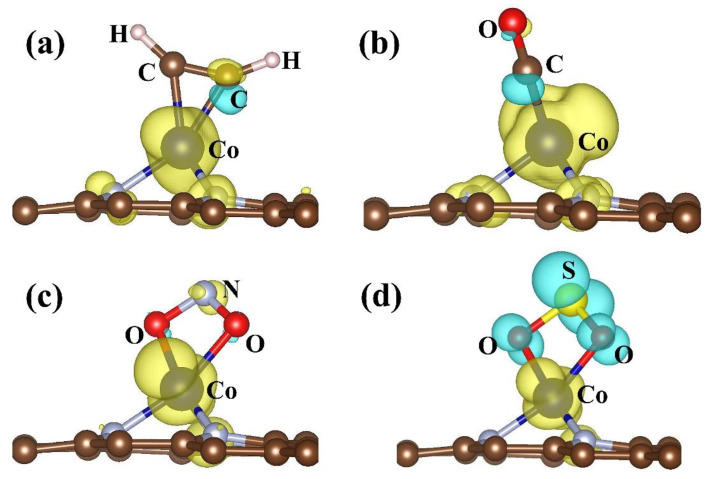
The spin density (Δρ = Δρ_↑_ − Δρ_↓_) of Co/N_3_–gra with (**a**) C_2_H_2_, (**b**) CO, (**c**) NO_2_, and (**d**) SO_2_ adsorbed on it. The yellow and blue areas indicate the positive and negative spin density respectively, isosurface value: 0.005 e/Bohr^3^.

**Figure 5 molecules-26-07700-f005:**
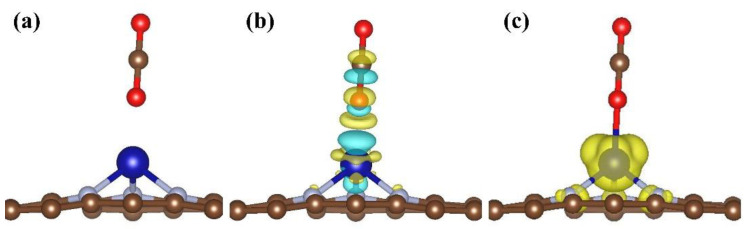
(**a**) The adsorption configuration of CO_2_ on Co/N_3_–gra; (**b**) The charge density difference of Co/N_3_–gra with CO_2_ adsorbed on it. The yellow and blue areas indicate the accumulation and depletion of charge respectively, isosurface value: 0.002 e/Bohr^3^. (**c**) The The spin density (Δρ = Δρ_↑_ − Δρ_↓_) of Co/N_3_–gra with CO_2_ adsorbed on it. The yellow and blue areas indicate the positive and negative spin density respectively, isosurface value: 0.005 e/Bohr^3^.

## Data Availability

All data generated or analysed during this study are included in this published article and its Appendix A.

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
