# Peer review of "The Adsorption Behavior of Gas Molecules on Co/N Co–Doped Graphene"

_molecules, 2021, doi:10.3390/molecules26247700_

Round 1
Reviewer 1 Report
The following comments are addressed to evaluate the work entitled: The adsorption behavior of gas molecules on Co/N co-doped graphene
In this work, the authors report the adsorption behavior of gas molecules on Co/N3 co-doped graphene (Co/N3-gra) via DFT. The geometric stability, electric properties, and magnetic properties are investigated upon the interaction between Co/N3-gra and gas molecules. The work can be accepted after a major revision.
The present work, in my opinion, has two kinds of drawbacks.
Minimal drawbacks:
- Labels in Figures (1, 2 and 4) are not readable, the authors should provide enhanced images with more readable labels. In the case of Figure 4, to maintain congruence, the authors should use the same labels (Top and Side) in the whole figure.
- The authors should corroborate that the references style is appropriate.
Major drawbacks:
- Nonetheless, authors have made a vast literature revision on this topic; there are fundamental contributions on this matter that should be considered. As a matter of fact, if other basic publications are considered, then this work would be markedly improved. For instance, the proposed system (Co/N) is well known in the literature. So, the authors should discuss similar contributions (https://doi.org/10.1016/j.molcata.2016.03.008 and https://doi.org/10.1016/j.apsusc.2020.146939) of the present work with respect to their work. Then, similar subject should be cited.
- The authors are invited to include recent data (https://doi.org/10.3390/s21061992 and https://doi.org/10.3390/ijms222312968) to give more insights into the Introduction section.
- The authors focused on graphene as a carbon allotrope without mentioning other usual carbon allotropes (https://doi.org/10.1016/j.matlet.2021.129660). The authors should consider comparing graphene with other common carbon allotropes to justify its use. Therefore, recent articles refereeing to a similar subject should be cited, and the authors should discuss the advantages/disadvantages of graphene with respect to other common carbon allotropes. It might be valuable for the readers to update the literature.
- The equation used to calculate the binding energy of single Co atom (Eb[Co]) in Co/N3-gra is hard to follow. The second term (E[d-Co/N3-gra]) is unclear.
Author Response
Minimal drawbacks:
- Labels in Figures (1, 2 and 4) are not readable, the authors should provide enhanced images with more readable labels. In the case of Figure 4, to maintain congruence, the authors should use the same labels (Top and Side) in the whole figure.
Thank you for reminding, we have corrected the figures in new styles.
- The authors should corroborate that the references style is appropriate.
Thank you for reminding, we have corrected the style of references into correct formation.
Major drawbacks:
- Nonetheless, authors have made a vast literature revision on this topic; there are fundamental contributions on this matter that should be considered. As a matter of fact, if other basic publications are considered, then this work would be markedly improved. For instance, the proposed system (Co/N) is well known in the literature. So, the authors should discuss similar contributions (https://doi.org/10.1016/j.molcata.2016.03.008 and https://doi.org/10.1016/j.apsusc.2020.146939) of the present work with respect to their work. Then, similar subject should be cited.
We thank the referree for the suggestion. We have added the discussion and cited the references.
This type of configuration has been widely investigated in pioneering works: Zhao et al. have embeded Ni atoms on different graphene-based materials as a novel material for detection of acid gases [14]. Yang et al. have also proposed a Co-N3 decorated graphene and applied it into CO oxidation [15].
- The authors are invited to include recent data (https://doi.org/10.3390/s21061992 and https://doi.org/10.3390/ijms222312968) to give more insights into the Introduction section.
We thank the referree for the suggestion. We have added the discussion and cited the references.
The metal decoration can effectively enhance the adsorption strenth of gas molecules: Co (Li)-decorated graphene can capture NO (CO) with -4.04 eV (-0.55 eV) adsorption energy [16].
- The authors focused on graphene as a carbon allotrope without mentioning other usual carbon allotropes (https://doi.org/10.1016/j.matlet.2021.129660). The authors should consider comparing graphene with other common carbon allotropes to justify its use. Therefore, recent articles refereeing to a similar subject should be cited, and the authors should discuss the advantages/disadvantages of graphene with respect to other common carbon allotropes. It might be valuable for the readers to update the literature.
We thank the referree for the suggestion. We have added the discussion and cited the references.
In recent years, many carbon allotropes, such as graphene and graphite, have been served as a magic material due to the perfect properties in various aspects including electricity, chemistry, mechanistic, thermology and optics [1,2]. Among these candidates, graphene is the most simple allotrope and has been extensively syntehsized in experimental works.
- The equation used to calculate the binding energy of single Co atom (Eb[Co]) in Co/N3-gra is hard to follow. The second term (E[d-Co/N3-gra]) is unclear.
We thank the referee for the constructive suggestion and we have revised the description more clearly.
|
Eb[Co] = E[Co] + E[v-Co/N3-gra] - E[Co/N3-gra] |
|
Where, the E[Co/N3-gra], E[v-Co/N3-gra], and E[Co] denote total energy of Co/N3-gra, Co/N3-gra with the Co atom moved away leaving as a Co vacancy (v-Co), and single Co atom in vacuum, respectively.

Reviewer 2 Report
The author of paper which title “The adsorption behavior of gas molecules on Co/N co-doped graphene”, evaluated the adsorption behavior of gas molecules on Co/N3 co-doped graphene (Co/N3-gra) using DFT calculation. The study is interesting, however it still requires enhancements before it can be accepted in the Molecules.
- The introduction part of the paper should be more in-details. Please highlight more previous literatures in the field and highlight the novelty of work.
- To highlight the goal of the paper, authors must compare the results of different gas molecules more in-details in different section of paper.
- Bringing the bandgap energy in the presence of different number of gas molecules and find out the stability of structure is necessary in this paper (at least 2 to 4 molecules).
- Authors should provide the DFT calculation results of CO2 in terms of energy density and compare to the CO to figure out the stability and semiconductor behavior of structure in the presence of oxygen molecule.
Author Response
- The introduction part of the paper should be more in-details. Please highlight more previous literatures in the field and highlight the novelty of work.
We thank the referee for the suggestion. We have added more description of previous leteratures to enrich the manuscript.
This type of configuration has been widely investigated in pioneering works: Zhao et al. have embeded Ni atoms on different graphene-based materials as a novel material for detection of acid gases [14]. Yang et al. have also proposed a Co-N3 decorated graphene and applied it into CO oxidation [15]. The metal decoration can effectively enhance the adsorption strenth of gas molecules: Co (Li)-decorated graphene can capture NO (CO) with -4.04 eV (-0.55 eV) adsorption energy [16].
Chen et al. have discovered the potential application of graphene as oxygen reduction catalyst and proposed the heteroatom doping as an excellent method to tune the catalytic reactivity [19]. Lukaszewicz et al. have also synthesized a micro-mesoporous graphene and highlighted the structural factors in application of Zn-air batteris [20].
- To highlight the goal of the paper, authors must compare the results of different gas molecules more in-details in different section of paper.
We thank the referee for the suggestion and we have added the comparison in the manuscript.
Above all, CO and C2H2 bind with Co-v-gra via C atoms while it is O atoms for NO2 and SO2. Besides, NO2 and SO2 exhibit enhanced binding strength compared to CO and C2H2 due to large charge transfer.
This tendency is surprisingly notable for NO2 and SO2, indicating the electron environment has changed dramatically around adsorption sites. CO can later the conductivity dramatically and C2H2 mianly change the magnetic properties. The mechanism is discussed in detail as follows.
- Bringing the bandgap energy in the presence of different number of gas molecules and find out the stability of structure is necessary in this paper (at least 2 to 4 molecules).
We thank the referee for the suggestion. The active center of our investigated systems possesses only one atom,which cannot adsorb more than one gas molecules simultaneously.
- Authors should provide the DFT calculation results of CO2 in terms of energy density and compare to the CO to figure out the stability and semiconductor behavior of structure in the presence of oxygen molecule.
The gas investigated is SO2 not CO2 and we write it as CO2 for mistake. We are sorry for the mistake and thank the referee for reminding.
About NO2 and SO2, the magnetic moment has decreased after adsorption on Co/N3-gra, which is different from that for C2H2 and CO. This may be due to the paramagnetic (nonmagnetic) property for NO2 (SO2) while the magnetic moment of Co/N3-gra is 2.0 μB in comparison [43]. The relative low magnetism of NO2 and SO2 can induce the simultaneous shift of Co/N3-gra’s states in up and down spin towards Fermi level, which eventually decrease the magnetic moment to 0.14~1.0 μB after adsorption of gas (Fig. 3c). The peaks in TDOS of Co/N3-gra have decreased after NO2/SO2 adsorption.

Reviewer 3 Report
Dear Authors
The manuscript entitled "The adsorption behavior of gas molecules on Co/N co-doped graphene" presented concerns an interesting and actual subject. This manuscript can be accepted after major revision. The following suggestion and comments should be taken:
- The authors could insert more numerical data into the Abstract for enhancement of the manuscript.
- The overall English needs to be improved. Please seek guidance from a native English speaker if possible (commas, plural form, "the" "a", and others could be corrected).
- (Line 26-29) Please add in the introduction more information about potential applications (medicine, electrochemistry for example batteries, supercapacitors, etc.) of high surface area graphene. Please cite: (1) The application of graphene and its composites in oxygen reduction electrocatalysis: a perspective and review of recent progress Energy Environ. Sci., 2016,9, 357-390 https://doi.org/10.1039/C5EE02474A, (2) High surface area micro-mesoporous graphene for electrochemical applications. Sci Rep 11, 22054 (2021). https://doi.org/10.1038/s41598-021-01154-0 and (3) DFT calculations of graphene monolayer in presence of Fe dopant and vacancy Physica B: Condensed Matter 541, (2018), 6-13, https://doi.org/10.1016/j.physb.2018.04.023.
- Line 87 (please put centre position)
- Why authors do not describe carbon ends saturated with hydrogen atoms in carbon materials calculations? Please explain in the comments. The dangling bonds of carbon ends may affect the structural and electronic properties of edge states.
- It is suggested to describe 2-3 sentences about DFT calculations for different size graphene systems to investigate the dependence on structure size of the graphene layer.
- Could the authors include the standard deviation of the used methods?
- Please, could the authors explain why they used such calculation methods and not others?
Thank you for the opportunity to review this manuscript.
Author Response
- The authors could insert more numerical data into the Abstract for enhancement of the manuscript.
We have added the numerical data in the abstract part.
The binding energy of Co is -5.13 eV, which is big enough for application in gas adsorption. For the adsorption of C2H4, CO, NO2, and SO2 on Co/N-gra, the molecules may act as donors or acceptors of electrons, which can lead to charge transfer (range from 0.38 to 0.7 e) and eventually change the conductivity of Co/N-gra.
- The overall English needs to be improved. Please seek guidance from a native English speaker if possible (commas, plural form, "the" "a", and others could be corrected).
We have polished the lauguage carefully and revised some grammar errors.
- (Line 26-29) Please add in the introduction more information about potential applications (medicine, electrochemistry for example batteries, supercapacitors, etc.) of high surface area graphene. Please cite: (1) The application of graphene and its composites in oxygen reduction electrocatalysis: a perspective and review of recent progress Energy Environ. Sci., 2016,9, 357-390 https://doi.org/10.1039/C5EE02474A, (2) High surface area micro-mesoporous graphene for electrochemical applications. Sci Rep 11, 22054 (2021). https://doi.org/10.1038/s41598-021-01154-0 and (3) DFT calculations of graphene monolayer in presence of Fe dopant and vacancy Physica B: Condensed Matter 541, (2018), 6-13, https://doi.org/10.1016/j.physb.2018.04.023.
We thank the referree for the suggestion. We have added the discussion and cited the references.
Chen et al. have discovered the potential application of graphene as oxygen reduction catalyst and proposed the heteroatom doping as an excellent method to tune the catalytic reactivity [19]. Lukaszewicz et al. have also synthesized a micro-mesoporous graphene and highlighted the structural factors in application of Zn-air batteris [20].
- Line 87 (please put centre position)
Thank you for reminding, we have put the equation into the centre position.
- Why authors do not describe carbon ends saturated with hydrogen atoms in carbon materials calculations? Please explain in the comments. The dangling bonds of carbon ends may affect the structural and electronic properties of edge states.
Thank you for reminding, the configurations we constructed are all periodic structures, which has no dangling bonds in the system. We have added the illustration in the methods part and corrected the style of fig.1.
The configurations we constructed are all periodic structures and the size of unit cell are set big enough to avoid the influence of Vander Waals force between molecules.
- It is suggested to describe 2-3 sentences about DFT calculations for different size graphene systems to investigate the dependence on structure size of the graphene layer.
Our systems are all periodic structures as replyed for question 5. Thus, the size should be big enough to avoid the influence of Vander Waals force between molecules. As the size of unit cell increased big enough, there exists little influence on the adsorption properties of molecules.
The configurations we constructed are all periodic structures and the size of unit cell are set big enough to avoid the influence of Vander Waals force between molecules.
- Could the authors include the standard deviation of the used methods?
The approximate deviation of DFT local functionals are 0.2 eV for energy calculation.
The approximate deviation of DFT local functionals are 0.2 eV for energy calculation, indicating the validation of DFT method in gas adsorption area.
- Please, could the authors explain why they used such calculation methods and not others?
DFT is an effective method in description of gas adsorption bahavior due to the ultralow error for energy calculation (0.2 eV) and has been widely used in pioneering works.
The approximate deviation of DFT local functionals are 0.2 eV for energy calculation, indicating the validation of DFT method in gas adsorption area.

Round 2
Reviewer 1 Report
Minimal drawbacks:
Page 2, line 49: "strenth" should be rewritten.
Author Response
We are sorry for the mistake and we have revised the error in the manuscript.
Reviewer 2 Report
The Authors of the Manuscript "The adsorption behavior of gas molecules on Co/N co-doped graphene" have revised the Manuscript according to the comments. The Manuscript has improved, but it still requires enhancements before it can be accepted in the Molecules.
In terms of the gas molecules, bringing the DFT results of CO2 compared to CO in terms of stability and energy density is necessary to have a better insight about Co/N co-doped graphene. If this work is done before please cite it if not please do this calculation and add to this paper.
Author Response
We have done some calculations about CO2 adsorption on Co/N co-doped graphene and added the corresponding discussions in the manuscript. For comparison, we have also calculated the CO2 adsorption. The CO2 exhibits phyiscal adsorption on Co/N3-gra with an adsorption energy of -0.28 eV (Fig. 5a and b). The bond angle of adsorbed CO2 maintains 180° as the free state. In addition, there exists no spin on CO2 adsorbed on Co/N3-gra (fig. 5c), indicating the adsorption of CO2 has nearly no change for the properties of the substrate. These phenomenon mainly originates from the neglegible charge transfer between CO2 and Co/N3-gra(only 0.08 e).

Reviewer 3 Report
Accept in present form.
Author Response
Thank you very much for agreeing to accept our manuscript.